# VOYAGER: REAL-TIME CITY-SCALE 3D GAUSSIAN SPLATTING ON RESOURCE-CONSTRAINED DEVICES

## ABSTRACT

3D Gaussian splatting (3DGS) is an emerging technique for photorealistic 3D scene rendering. However, rendering city-scale 3DGS scenes on resource-constrained mobile devices in real-time remains a significant challenge due to two compute-intensive stages: level-of-detail (LoD) search and rasterization.

In this paper, we propose VOYAGER, an effective solution to accelerate city-scale 3DGS rendering on mobile devices. Our key insight is that, *under normal user motion, the number of newly visible Gaussians within the view frustum remains roughly constant*. Leveraging this temporal correlation, we propose a temporal-aware LoD search to identify the necessary Gaussians for the remaining rendering stages. For the remaining rendering process, we accelerate the bottleneck stage, rasterization, via preemptive $\alpha$-filtering. With all optimizations above, our system can deliver low-latency, city-scale 3DGS rendering on mobile devices. Compared to existing solutions, VOYAGER achieves up to $6.6\times$ speedup and 85% energy savings with superior rendering quality. Codes will be released upon publication.

## 1 INTRODUCTION

The field of 3D Gaussian splatting (3DGS) (Kerbl et al., 2023; Fan et al., 2024; Fang & Wang, 2024; Girish et al., 2024; Lee et al., 2024a; Niemeyer et al., 2024; Wang et al., 2024a; Mallick et al., 2024; Gui et al., 2024; Radl et al., 2024) has made tremendous progress in recreating the 3D world with photo-realistic fidelity. While 3DGS unlocks new possibilities and fuels our imagination, rendering city-scale 3DGS scenes on mobile devices, e.g., the smartphone in your pocket, remains a substantial challenge. Primarily, there are some inherent limitations of mobile devices, such as limited compute resources and memory bandwidth. For example, a single scene like *SmallCity* from HierarchicalGS (Kerbl et al., 2024) can barely achieve 12.2 frame-per-second (FPS) on a Nvidia mobile Ampere GPU (Nvidia, 2023). These constraints make it infeasible to achieve city-scale 3DGS on today's mobile devices.

In order to render massive-scale scenes, 3DGS algorithms require managing tens of millions of Gaussian ellipsoids, a.k.a. Gaussians. For instance, *SmallCity* from HierarchicalGS (Kerbl et al., 2024) contains over $1.8 \times 10^7$ Gaussians. To effectively manage at this scale, the state-of-the-art city-scale 3DGS methods (Kerbl et al., 2024; Ren et al., 2024; Liu et al., 2024) often adopt a two-stage pipeline: *level-of-detail (LoD) search* and *splatting*. LoD search uses a hierarchical representation to manage those Gaussians and selects a subset of Gaussians at an appropriate LoD under the current camera pose. Splatting then renders these selected Gaussians on the canvas in a visually consistent order. Our experiment in Fig. 2 shows that both stages take non-trivial execution time, with the LoD search further dominating as the scene complexity increases.

To reduce the computation workload of LoD search, our key insight is that, as the user navigates through a 3D scene, *the number of newly visible Gaussians introduced by continuous viewpoint changes remains roughly constant* (assuming a general case of a smooth user motion). The subsets of selected Gaussians across adjacent frames are often similar in LoD search. By leveraging this insight, we propose a *temporal-aware LoD search* algorithm in Sec. 4 .1. Specifically, we propose two GPU-oriented optimizations in LoD search. For the initial frame, we propose an efficient tree traversal algorithm that can be highly parallelized on GPUs to accelerate LoD search. For the subsequent frames, we design a companion data structure that leverages temporal correlations across consecutive tree traversals to further accelerate this process. Rather than recomputing the tree traversal

results from scratch, our data structure can reuse previous traversal results to reduce the computation. Compared against the previous LoD search algorithms (Kerbl et al., 2024), we achieve 6.5× speedup with the bit-accurate LoD search result.

To improve the efficiency of splatting, prior studies (Wang et al., 2024a; Fan et al., 2024; Fang & Wang, 2024; Huang et al., 2025; Lin et al., 2025) have primarily focused on reducing the total number of Gaussians in rasterization, a sub-step in splatting. However, these methods often overlook the underlying bottleneck within rasterization itself. We pinpoint the major bottleneck in rasterization and propose a GPU-oriented optimization, *preemptive $\alpha$-filtering* in Sec. 4 .2, to accelerate rasterization. Our key observation is that the primary source of slowdown during rasterization stems from the per-ray computation of Gaussian transmittance, $\alpha_i$. This operation is computationally expensive due to the heterogeneous design of modern GPU architectures (see Sec. 3 .2). To address this issue, our filter can pre-determine the subset of Gaussians that do not need to compute transmittance, $\alpha_i$, without explicitly calculating $\alpha_i$. For those Gaussians that do require $\alpha$-computation, we bypass the costly $\alpha$-computation by embedding a 128-byte precomputed lookup table (LUT) in the GPU's shared memory and approximating Gaussian transmittance instead.

Together, our algorithm can achieve up to 6.6× speedup compared to prior city-scale 3DGS algorithms on a mobile platform, Nvidia Orin SoC. The contributions of this paper are as follows:

- A parallel LoD search algorithm that exploits temporal coherence across frames to avoid repetitive tree traversal in LoD search.
- A GPU-efficient rasterization method that bypasses the key bottleneck in rasterization and significantly improves rendering efficiency.
- A framework that enables rendering city-scale 3DGS models on mobile devices, achieving up to 6.6× speedup and 85% energy savings compared to prior works.

## 2 RELATED WORK

**Efficient 3DGS.** State-of-the-art methods for the efficiency of 3DGS can largely be categorized into two main directions. The first direction focuses on reducing model size. Several studies propose various pruning techniques (Fan et al., 2024; Fang & Wang, 2024; Girish et al., 2024; Niemeyer et al., 2024; Feng et al., 2024b; Peng et al., 2024; Tu et al., 2024; Franke et al., 2024; Hahlbohm et al., 2025), while others introduce alternative representations to minimize storage requirements (Roessle et al., 2024; Lee et al., 2024a; Huang et al., 2024; Wang et al., 2024b; Dai et al., 2024; Lee et al., 2024a; Girish et al., 2024).

The second direction aims to improve the efficiency of the 3DGS rendering pipeline. Some approaches introduce fine-grained filtering techniques for Gaussian-ray intersections to alleviate the computation of subsequent stages (Feng et al., 2024a; Lee et al., 2024b; Wang et al., 2024a). For instance, various online filtering techniques, such as axis-aligned bounding box (AABB) (Wang et al., 2024a) and oriented bounding box (OBB) (Lee et al., 2024b) intersection tests, have been proposed for more fine-grained filtering. Others tackle GPU performance bottlenecks such as warp divergence and workload imbalance in rasterization (Huang et al., 2025; Gui et al., 2024; Mallick et al., 2024).

In contrast to these prior works, our framework focuses on large-scale 3DGS rendering. We jointly optimize both the LoD search and splatting stages, while addressing the performance constraints and hardware limitations that exist in today's mobile GPUs.

**Large-scale 3DGS.** How to represent Level-of-Detail (LoD) efficiently is an issue existed broadly in neural rendering (Turki et al., 2023; Xu et al., 2023; Barron et al., 2023). Since 3DGS models are inherently point-based representations, various tree-based structures have been designed to manage large-scale scenes, such as Octree-GS (Ren et al., 2024), CityGaussian (Liu et al., 2024), and HierarchicalGS (Kerbl et al., 2024). These hierarchical structures organize Gaussian points to support scalable rendering. However, the irregular tree traversals introduce non-trivial runtime overhead.

In contrast, our LoD search is co-designed with GPU execution characteristics, such as the SIMT execution model. With the temporal correlation that commonly exists in continuous rendering, our method can achieve more efficient tree traversal and improve the overall runtime performance.

Fig. 1: Overview of rendering pipeline for large-scale 3D Gaussians. LoD search consists of two sub-steps: cut finding and weight interpolation. Splatting contains three sub-steps: projection, sorting, and rasterization. LoD search traverses the LoD tree to determine a set of Gaussians, given a LoD granularity. The result Gaussians form a "cut" that separates the top and bottom of the LOD tree. Then, the Gaussians on the cut go through a sequence of operations to render an image.

## 3 MOTIVATION

In this section, we first introduce the general pipeline of the large-scale 3DGS algorithm in Sec. 3 .1. We then analyze the performance bottlenecks of existing algorithms on modern GPUs in Sec. 3 .2.

### 3 .1 PRELIMINARY

**LoD Tree.** All city-scale 3DGS algorithms require a hierarchical tree-like structure to represent the entire scene. This data structure serves two main purposes: first, it identifies the Gaussians inside the view frustum; second, it supports rendering at an appropriate level of detail. Here, we describe the most general form of this data structure, *LoD tree*, where each tree level represents a specific detail granularity (see Fig. 1). Each tree node contains a single Gaussian with an unfixed number of child nodes. Gaussians in lower levels are generally smaller and provide finer details.

**Pipeline.** Fig. 1 gives an overview of the pipeline, which consists of two main stages: *LoD search* and *splatting*. Each of these stages includes several sub-steps, which we describe as follows:

- *Cut Finding*: This step traverses the LoD tree from top to bottom. At each node, we assess if the projected dimension of the Gaussian is smaller than the predefined LoD ($\tau^*$), while the projected dimension of its parent node is larger. We then gather all the Gaussians that meet this criterion. Essentially, these Gaussians form a "cut" that separates the top and bottom of the LoD tree, as shown in Fig. 1.

- *Weight Interpolation*: Each selected Gaussian then interpolates with its parent node to ensure a smooth transition across different LoDs. The interpolated Gaussians are inserted into the rendering queue in Fig. 1.

- *Culling & Projection*: Once the cut is determined, the selected Gaussians are projected onto the image plane. Gaussians outside the view frustum or deemed irrelevant are culled.

- *Sorting*: The remaining Gaussians are then sorted by depth, from the nearest to the farthest.

- *Rasterization*: The final step contains two parts. First, each pixel calculates the intersected transmittance $\alpha_i$ for each Gaussian. If $\alpha_i$ is below a predefined threshold, $\alpha^*$, the Gaussian is skipped for color blending. Otherwise, the Gaussian contributes to the final pixel color via weighted blending.

### 3 .2 PERFORMANCE ANALYSIS

**Execution Breakdown.** Fig. 2 shows the execution breakdown of HierarchicalGS across various LoDs on a Nvidia mobile Ampere GPU (Nvidia, 2023). Here, we use the *SmallCity* scene from the HierarchicalGS dataset as an example. When the LoD ($\tau^*$) is low, execution time is dominated by the rasterization step in the splatting stage. As the LoD increases, the cut finding step in the LoD search stage becomes the main contributor to overall execution time, accounting for up to 67%. On average, cut finding and rasterization accounts for 83% of total execution time. This means that both steps must be accelerated to achieve substantial speedups by Amdahl's Law (Amdahl, 1967).

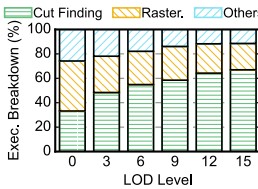 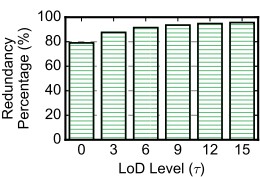 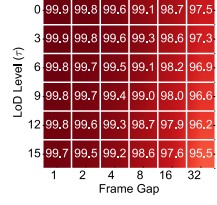 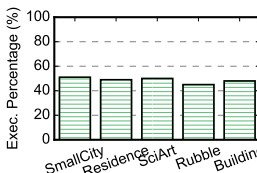

Fig. 2: Normalized execution breakdown across different LoDs.

Fig. 3: The percentage of redundant tree node accesses across LoDs.

Fig. 4: The overlap of selected Gaussians in the cuts between frames.

Fig. 5: The execution percentage of $\alpha$ computation in rasterization.

**Cut Finding.** We identify two primary inefficiencies in cut finding. First, existing methods primarily rely on exhaustive search to find the cut, as the inherent irregularity of the LoD tree makes it difficult to achieve efficient parallelism on modern GPUs. Such a brute-force approach often leads to substantial redundant computation. Our experiment shows that 92% of tree node accesses are unnecessary when the LoD is 6 (see Fig. 3). Second, in real-time rendering scenarios, repeatedly performing tree traversal from the root node for every frame introduces considerable redundant computation. We simulate a 60 FPS real-time rendering scenario and analyze the overlap in selected Gaussians across consecutive frames. Fig. 4 shows that 99% of the selected Gaussians remain unchanged across frames when the LoD is 6. This indicates that a substantial amount of computation is wasted due to repetitive tree traversal.

**Rasterization.** While a few studies have addressed certain inefficiencies in rasterization, such as taming workload imbalance (Wang et al., 2024a) and reducing warp divergence (Huang et al., 2025), existing works have largely overlooked the most time-consuming operation in rasterization: computing the Gaussian transmittance, $\alpha_i$, for each pixel-Gaussian pair. Using Nvidia Nsight Compute (Nvidia, 2025), Fig. 5 shows that this operation takes roughly 50% of the total rasterization time. The root cause is the imbalanced compute units within modern GPUs. Taking Nvidia GPUs as an example, most operations can be massively parallelized by CUDA cores. However, the computation of Gaussian transmittance requires exponential computation, which is executed by special functional units (SFUs). Since CUDA cores vastly outnumber the SFUs (16:1) (Krashinsky et al., 2020), calculating Gaussian transmittance becomes a key bottleneck.

## 4 METHODOLOGY

To address the challenges in Sec. 3.2, we present our framework, VOYAGER, which introduces two key optimizations, temporal-aware LoD search (Sec. 4.1) and preemptive $\alpha$-filtering (Sec. 4.2), targeting cut finding and rasterization in city-scale 3DGS rendering, respectively.

### 4.1 TEMPORAL-AWARE LOD SEARCH

We first explain our data structure, which is augmented on top of the original LoD tree to enhance GPU parallelism. Next, we explain our fully-streaming LoD tree traversal that can streamingly process subtrees in parallel on off-the-shelf GPUs. Finally, we describe how our temporal-aware LoD search can leverage previous cut results to further accelerate the LoD search.

**LoD Tree Partitioning.** Recall, from Sec. 3.1, each tree node in the LoD tree contains a single Gaussian with a variable number of child nodes. An important performance issue with this irregular data structure is the workload imbalance across different GPU warps[1]. To address this issue, we propose to partition the LoD tree into smaller chunks, such that each chunk has a similar workload.

In our partitioning algorithm, we first split the entire LoD tree into $N$ small subtrees from the top to the bottom. Each subtree is constrained to have a maximal tree height of 2. Once the decomposition is complete, the resulting subtrees are sorted in a breadth-first search (BFS) order and stored in a

---

[1] A GPU warp is a group of threads that execute the same instruction in lockstep.

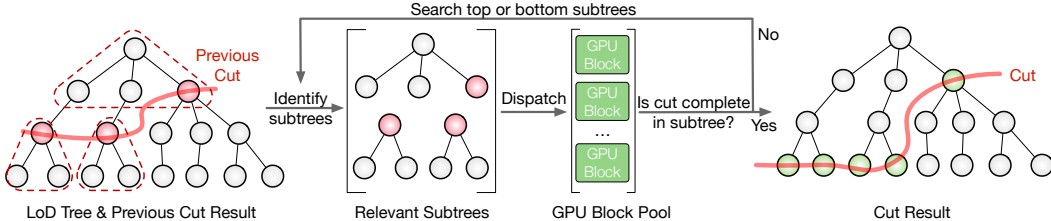

Fig. 6: The overview of temporal-aware LoD search. Our algorithm exploits the temporal correlations across frames and reuses the previous "cut" result to avoid the redundant tree traversal.

list, $\mathbb{S} = \{\mathcal{S}_0, \mathcal{S}_1, ..., \mathcal{S}_N\}$, where each $\mathcal{S}_i$ represents a single subtree. With this transformation, the tree partitioning problem can be converted into a constrained bin packing problem (Martello & Toth, 1990). The objective is to pack the subtrees into the minimum number of chunks, each corresponding to a batch of work for one GPU block. The constraint is that each chunk must contain subtrees with contiguous IDs in $\mathbb{S}$, to ensure the entire tree is traversed from the top to bottom.

*Optimization Objective.* Given this setup, we can convert this bin packing problem into a linear constrained optimization. The objective is to minimize the number of chunks:

$$\underset{\Theta}{\operatorname{argmin}} \mathcal{F}(\Theta) = \operatorname{Size}(\Theta), \text{ where } \Theta = \{\theta_i\}, \ \theta_i \in \{0, 1\}, \text{ and } i \in [0, 1, ..., N-1]. \quad (1)$$

Here, each non-zero $\theta_i$ represents the starting index of a new chunk. All non-zero $\theta_i$ are grouped into $\mathbb{I}$. Any two consecutive non-zero entries, $\theta_n$ and $\theta_m$, define a chunk that includes the subtree IDs within $[n, m)$. These subtrees are grouped and assigned to a single GPU block.

*Memory Constraints.* To maximize usage of the on-chip memory on GPUs, we set the total size of each chunk to be smaller than the GPU shared memory size ($M$) per streaming multiprocessor,

$$\forall \theta_m \in \mathbb{I}, \ \sum_{\theta_i = \theta_m}^{\theta_{m+1}-1} \operatorname{SubtreeSize}(\theta_i) \leq M. \quad (2)$$

This design ensures that all traversal operations for a chunk can be performed entirely within shared memory, avoiding off-chip memory stalls that would otherwise stall the GPU pipeline. Given this formulation, we can easily apply existing integer linear programming solvers, such as Google OR-Tools (Google, 2025), to obtain the optimal LoD tree partitioning offline.

**Fully-Streaming LoD Tree Traversal.** With our LoD tree partitioning, we propose a fully-streaming algorithm that achieves high parallelism on GPUs while avoiding unnecessary tree node visits. Rather than relying on the inherent parent-child relations in the original LoD tree, we perform tree traversal chunk by chunk in BFS order. During the GPU execution, each chunk is assigned to a GPU warp. The tree nodes within one chunk are then evenly distributed among threads within a warp to ensure a balanced workload across GPU threads. Since each chunk of nodes is small enough to fit in GPU shared memory, our algorithm streamingly processes those tree nodes and avoids irregular global memory accesses in the original LoD tree traversal.

The workload assignment is dynamically dispatched whenever a warp becomes available at runtime. The tree traversal terminates once a clean cut separates the top and bottom of the LoD tree, thereby avoiding redundant tree accesses. Meanwhile, we adopt the warp specialization (Bauer et al., 2014) that uses a small group of dedicated GPU threads within each warp for data loading. This allows data loading to be overlapped with computation, hiding the latency of loading LoD tree chunks.

**Temporal-Aware LoD Search.** For tree traversal of subsequent frames, we further exploit the temporal correlations across frames and introduce a temporal-aware LoD search. Our algorithm also leverages our LoD tree partitioning and works on the subtree traversal. Fig. 6 highlights some individual subtrees in red dashed blocks. Here, we only show a two-level subtree partitioning for illustration purposes; our actual implementation applies multi-level partitioning to the LoD tree.

The overall algorithm is also described in Algo. 1. Given the cut result, $\mathbb{R}_{prev}$, from the previous frame (highlighted in pink in Fig. 6), our algorithm first identifies the subtrees, $\mathbb{S}_{prev}$, to which each

---

**Algorithm 1:** Algorithm of Temporal-Aware LoD Search

---

**Data:** LoD tree $\mathbb{S}$, Previous cut, $\mathbb{R}_{prev}$, LoD granularity, $\tau^*$.
**Result:** cut $\mathbb{R}_{cut}$.
$\mathbb{S}_{prev} \leftarrow$ identifySubtreeInPrevCut($\mathbb{S}, \mathbb{R}_{prev}, \tau^*$);
**while** $\mathbb{S}_{prev}$ *is not empty* **do**
  $\mathcal{S}_i \leftarrow \mathbb{S}_{prev}$.dequeue();
  $F_{signal}, \mathbb{R}_{tmp} \leftarrow$ findCut($\mathcal{S}_i, \tau^*$); // delegate to one GPU warp if available;
  **for** $n_i$ *in* $\mathbb{R}_{tmp}$ **do**
    $\mathbb{R}_{cut}$.enqueue($n_i$); // enqueue the subtree result;
  **end**
  **if** $F_{signal}$ == *TOP and* $\mathbb{S}_{prev}$.*parent is not visited* **then**
    $\mathbb{S}_{prev}$.push($\mathbb{S}_{prev}$.parent); // too fine-grained, need to find its parent;
  **end**
  **if** $F_{signal}$ == *BOTTOM* **then**
    **for** $n_i$ *in* $\mathbb{R}_{tmp}$ **do**
      $\mathbb{S}_{prev}$.push(); // too coarse-grained, need to find its children;
    **end**
  **end**
**end**

---

Gaussian in the cut belongs. Then, our algorithm only traverses those identified subtrees instead of all subtrees in a GPU streaming fashion. In GPU implementation, each subtree is assigned to a separate GPU warp for local subtree traversal. Because each subtree is approximately equal in size, it ensures a balanced workload distribution across GPU warps. If searching the local subtree cannot obtain the complete "cut" result, i.e., no clean cut for this subtree (signaled by $F_{signal}$), we then search its corresponding parent subtree or child subtrees to complete the cut finding. This way, we ensure that the results from our temporal-aware LoD search are bit-accurate.

### 4.2 PREEMPTIVE $\alpha$-FILTERING

In Sec. 3.2, we show that the primary performance bottleneck in rasterization is the computation of Gaussian transmittance, which accounts for over 50% of the total execution time. This operation introduces significant latency because it relies on exponential computations. Unlike common arithmetic operations, exponentials are not executed by the massive CUDA cores. Instead, they are executed by SFUs, which are far fewer in number compared to CUDA cores (1:16). This imbalance in hardware resources makes transmittance computation a major bottleneck in rasterization.

To address this, we propose *preemptive $\alpha$-filtering* to accelerate rasterization. In canonical rasterization, the Gaussian transmittance, $\alpha$, is computed by,

$$\alpha = \min(0.99, \theta_i \cdot e^\rho), \tag{3}$$

where $\rho$ is the power decay of the current Gaussian-ray intersection and $\theta_i$ is the constant opacity of the intersected Gaussian $i$. Each $\alpha_i$ is then compared against a transmittance threshold $\alpha^*$ to determine whether to proceed with the subsequent color blending.

Instead of computing the expensive exponential for every Gaussian-ray intersection, we refactor the transmittance comparison as,

$$\log(\alpha^*) - \log(\theta_i) < \rho. \tag{4}$$

Since $\log(\theta_i)$ can be precomputed offline and stored for each Gaussian, we replace the original transmittance check with our lightweight check, $\log(\alpha^*) - \log(\theta_i) < \rho$, before computing the transmittance, $\theta_i \cdot e^\rho$. This optimization has two advantages. First, it allows us to skip unnecessary exponent computations for Gaussians that are below the transmittance threshold, $\alpha^*$. Second, by moving this preemptive check to the *culling&projection* stage, we can further reduce the computation overheads for subsequent stages: sorting and rasterization.

Although our lightweight check reduces the number of exponential operations, it cannot eliminate them completely. To remove these computations altogether, we introduce a lookup table (LUT)

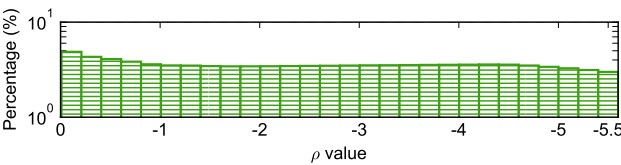

Fig. 7: The distribution of $\rho$ values across the value space, $[-5.55, 0]$.

Table 1: Quantitative evaluation of VOYAGER against the state-of-the-arts (Kerbl et al., 2023; Liu et al., 2024; Ren et al., 2024; Kerbl et al., 2024). Here, we show the results on three large-scale datasets: HierarchicalGS (Kerbl et al., 2024), UrbanScene3D (Lin et al., 2022), and MegaN-eRF (Turki et al., 2022). We highlight the **best** and second-best results among all methods.

| Dataset | HierarchicalGS | | | | | | UrbanScene3D | | | | | | MegaNeRF | | | | | |
|---|---|---|---|---|---|---|---|---|---|---|---|---|---|---|---|---|---|---|
| | Quality | | | Efficiency | | | Quality | | | Efficiency | | | Quality | | | Efficiency | | |
| Metrics | PSNR↑ (dB) | SSIM↑ | LPIPS↓ | FPS↑ (Orin) | #Op.↓ ($10^6$) | Energy↓ (mJ) | PSNR↑ (dB) | SSIM↑ | LPIPS↓ | FPS↑ (Orin) | #Op.↓ ($10^6$) | Energy↓ (mJ) | PSNR↑ (dB) | SSIM↑ | LPIPS↓ | FPS↑ (Orin) | #Op.↓ ($10^6$) | Energy↓ (mJ) |
| 3DGS | 23.32 | 0.724 | 0.430 | 12.18 | 732.69 | 443.52 | 21.65 | 0.818 | 0.269 | 11.60 | 2544.25 | 479.88 | 22.96 | 0.752 | 0.330 | 12.03 | 1630.92 | 415.46 |
| CityGaussian | 22.01 | 0.729 | 0.344 | 24.86 | 649.68 | 209.16 | 21.73 | 0.825 | 0.223 | 19.24 | 2522.30 | 276.16 | 24.27 | 0.797 | 0.238 | 20.40 | 2326.09 | 261.98 |
| OctreeGS | 21.33 | 0.646 | 0.415 | 13.55 | 2068.59 | 398.54 | 21.97 | 0.802 | 0.247 | 30.30 | 2121.52 | 186.43 | 24.56 | 0.765 | 0.272 | 29.04 | 1198.70 | 176.87 |
| HierarchicalGS ($\tau = 3.0$) | 26.24 | 0.812 | 0.254 | 11.43 | 857.64 | 455.03 | 23.79 | 0.755 | 0.228 | 12.69 | 2502.38 | 409.96 | 25.18 | 0.783 | 0.280 | 15.39 | 1628.42 | 344.55 |
| HierarchicalGS ($\tau = 15.0$) | 25.39 | 0.771 | 0.316 | 15.53 | 591.01 | 334.73 | 23.11 | 0.716 | 0.306 | 17.94 | 1547.26 | 290.29 | 23.31 | 0.686 | 0.367 | 19.63 | 1182.06 | 270.57 |
| VOYAGER ($\tau = 3.0$) | 26.24 | 0.812 | 0.254 | 42.19 | 584.15 | 127.99 | 23.78 | 0.755 | 0.228 | 39.01 | 1118.84 | 139.69 | 25.17 | 0.783 | 0.280 | 56.42 | 528.81 | 99.12 |
| VOYAGER ($\tau = 15.0$) | 25.39 | 0.771 | 0.316 | 68.84 | 363.68 | 78.44 | 23.11 | 0.716 | 0.306 | 58.67 | 653.54 | 92.05 | 23.31 | 0.686 | 0.367 | 79.38 | 362.02 | 70.60 |

embedded in GPU shared memory. The LUT can directly map $\rho$ values to the corresponding exponential results, $e^\rho$, allowing us to obtain $e^\rho$ via a memory lookup rather than invoking SFUs.

The next question is how to design such a LUT. Our observation is that $\rho$ values are uniformly distributed between the effective value space, $[-5.55, 0]$, as shown in Fig. 7. Thus, we uniformly split the effective value space into $m$ intervals. Here, we choose m to be 32. a given $\rho$ value can be directly indexed to this LUT by computing,

$$\text{index} = \left\lfloor \frac{\rho}{-5.55} \cdot m \right\rceil, \tag{5}$$

where $\lfloor \cdot \rceil$ denotes rounding to the nearest integer. The LUT indexing enables low-latency exponential approximation via a small shared memory. The sensitivity of $m$ is shown in Fig. 10.

## 5 EVALUATION

### 5.1 EXPERIMENTAL SETUP

To show the efficiency and robustness of VOYAGER, we evaluate on three large-scale datasets: HierarchicalGS (Kerbl et al., 2024), UrbanScene3D (Lin et al., 2022), and MegaNeRF (Turki et al., 2022). Meanwhile, we also test VOYAGER on small-scale datasets: Mip-NeRF360 (Barron et al., 2021), Tank&Temple (Knapitsch et al., 2017), and DeepBlending (Hedman et al., 2018). For quality evaluation, we use PSNR, SSIM, and LPIPS. We also report frame-per-second (FPS), the number of executed operations (#Op.), and energy consumption as our performance metrics.

To evaluate the performance, our hardware platform is the mobile Ampere GPU on Nvidia AGX Orin SoC with 5.33 TFLOPs, which is the flagship development board in AR/VR. For GPU performance, we measure latency, including the execution time as well as the kernel launch on the mobile Ampere GPU. The GPU power is directly obtained using the built-in power sensing circuitry.

For comparison, we compare against three large-scale 3DGS algorithms: HierarchicalGS (Kerbl et al., 2024), CityGaussian (Liu et al., 2024), and Octree-GS (Ren et al., 2024). We also compare against a dense 3DGS algorithm: 3DGS (Kerbl et al., 2023). In our evaluation, our algorithm VOYAGER applies optimizations in LOD search and splatting.

### 5.2 RENDERING QUALITY ON LARGE-SCALE DATASETS

Tbl. 1 shows the overall performance and quality comparison between VOYAGER and other baselines. Across three quality metrics, both HierarchicalGS and VOYAGER with $\tau$ of 3 achieve the

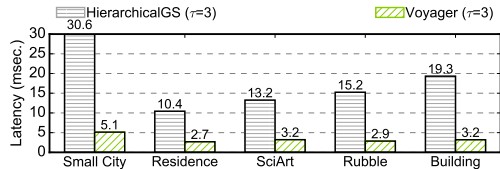 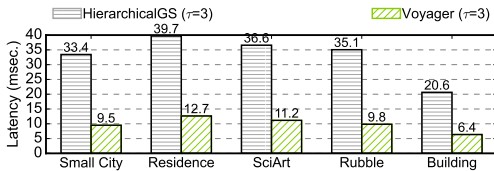

Fig. 8: The standalone latency comparison of LoD search between HierarchicalGS and VOYAGER with $\tau$ of 3. We accelerate 5.1× speedup compared to HierarchicalGS.

Fig. 9: The standalone latency comparison of splatting between HierarchicalGS and VOYAGER with $\tau$ of 3. We accelerate 3.3× speedup compared to HierarchicalGS.

Table 2: Ablation Study of individual contributions in VOYAGER on the HierarchicalGS dataset. FS: Fully-streaming LoD search. Temporal: Temporal-aware LoD search. FPS is measured on Orin.

| Metrics | Quality | | | Efficiency | | |
|---|---|---|---|---|---|---|
| | PSNR (dB)↑ | SSIM↑ | LPIPS↓ | FPS↑ | #Op. ($10^6$)↓ | Energy (mJ)↓ |
| Base ($\tau = 3.0$) | 26.24 | 0.812 | 0.254 | 11.43 | 857.64 | 455.03 |
| Base+FS ($\tau = 3.0$) | 26.24 | 0.812 | 0.254 | 15.78 | 746.01 | 335.87 |
| Base+Temporal ($\tau = 3.0$) | 26.24 | 0.812 | 0.254 | 22.15 | 681.63 | 239.12 |
| VOYAGER ($\tau = 3.0$) | 26.24 | 0.812 | 0.254 | 42.19 | 584.15 | 127.99 |

overall highest quality. These two achieve similar scores on quality metrics, because the LUT-based exponential approximation in Sec. 4 .2 introduces negligible accuracy loss. Other optimizations introduced by VOYAGER all produce the bit-accurate results as HierarchicalGS. In comparison, dense models, e.g., 3DGS, perform poorly on large-scale scenes due to their lack of a mechanism to manage LoD. More qualitative comparisons are shown in the appendix.

## 5 .3 RENDERING EFFICIENCY

**Speedup.** In terms of rendering speed, VOYAGER achieves the highest FPS among all evaluated methods in Tbl. 1. Specifically, VOYAGER can achieve up to 68.84, 58.67, and 79.38 FPS on three large-scale datasets, respectively. It is also the only method capable of delivering real-time rendering (≥60 FPS) consistently across all tested scenes. Compared to dense models such as 3DGS, VOYAGER delivers up to 6.6× speedup with better rendering quality. Compared to the corresponding large-scale 3DGS algorithm, HierarchicalGS, VOYAGER still achieves 4.0× speedup. Fig. 8 and Fig. 9 further dissect our speedup contribution and show the standalone speedup on LoD search and rasterization, respectively. Compared to HierarchicalGS, VOYAGER achieves 5.1× speedup on LoD search and 3.3× speedup on splatting (including projection, sorting, and rasterization).

In terms of total executed operations (#Op.), VOYAGER achieves the lowest computational cost. This efficiency is primarily attributed to two key optimizations. First, the temporal-aware LoD search reduces redundant tree node accesses. Second, the preemptive $\alpha$-filtering shifts a transparency check to the early projection stage and naturally reduces the workload of the subsequent stages. Lastly, the overall trend in energy savings follows the speedup results. VOYAGER achieves the highest energy savings with 82% and 80% of energy savings compared to 3DGS and HierarchicalGS, repetively.

## 5 .4 ABLATION STUDY

Tbl. 2 presents an ablation study that breaks down the contributions of individual components in VOYAGER. Because the only approximation introduced by VOYAGER is the LUT-based exponential approximation, it only introduces a negligible quality loss. All other acceleration techniques produce lossless and bit-accurate results. From an efficiency perspective, each component contributes to the overall speedup. Our fully streaming LoD search can achieve 1.4× speedup. On top of that, our temporal-aware LoD search can further boost the overall performance to 1.9×. Due to Amdahl's Law, only optimizing LoD search would not achieve a significant speedup. By further accelerating the splatting stage via our preemptive $\alpha$-flitering, VOYAGER eventually boosts the performance to 3.7× on the HierarchicalGS dataset.

Table 3: Evaluation of VOYAGER against the state-of-the-arts on three small-scale datasets: Mip-NeRF360 (Barron et al., 2021), Tank&Temple (Knapitsch et al., 2017), DeepBlending (Hedman et al., 2018). We highlight the **best** and second-best results among all methods. *Note that, rendering small-scale scenes is **not** the primary focus of this work; we include the results for completeness.*

| Dataset | MipNeRF360 | | | | | Tank&Temple | | | | | DeepBlending | | | | |
|---|---|---|---|---|---|---|---|---|---|---|---|---|---|---|---|
| Metrics | PSNR↑ (dB) | Quality | | Efficiency | | PSNR↑ (dB) | Quality | | Efficiency | | PSNR↑ (dB) | Quality | | Efficiency | |
| | | SSIM↑ | LPIPS↓ | FPS↑ (Orin) | Energy↓ (mJ) | | SSIM↑ | LPIPS↓ | FPS↑ (Orin) | Energy↓ (mJ) | | SSIM↑ | LPIPS↓ | FPS↑ (Orin) | Energy↓ (mJ) |
| 3DGS | 27.53 | 0.815 | 0.220 | 13.07 | 428.79 | 23.76 | 0.852 | 0.169 | 23.95 | 215.21 | 29.80 | 0.907 | 0.238 | 14.73 | 368.21 |
| CityGaussian | 27.50 | 0.813 | 0.221 | 14.42 | 380.82 | 23.71 | 0.848 | 0.177 | 27.20 | 193.11 | 29.54 | 0.904 | 0.244 | 14.63 | 382.83 |
| OctreeGS | 27.80 | 0.846 | 0.198 | 20.16 | 362.75 | 24.39 | 0.856 | 0.170 | 36.31 | 135.10 | 29.92 | 0.903 | 0.262 | 26.67 | 141.18 |
| HierarchicalGS ($\tau$ = 3.0) | 27.24 | 0.818 | 0.213 | 16.44 | 320.30 | 25.65 | 0.850 | 0.170 | 24.93 | 194.06 | 30.29 | 0.904 | 0.260 | 26.95 | 203.76 |
| HierarchicalGS ($\tau$ = 15.0) | 24.71 | 0.718 | 0.294 | 16.89 | 308.00 | 24.72 | 0.819 | 0.233 | 31.95 | 149.19 | 29.02 | 0.892 | 0.277 | 28.71 | 191.97 |
| VOYAGER ($\tau$ = 3.0) | 27.24 | 0.818 | 0.213 | 58.10 | 110.38 | 25.65 | 0.850 | 0.170 | 46.52 | 134.30 | 30.29 | 0.904 | 0.260 | 53.27 | 131.44 |
| VOYAGER ($\tau$ = 15.0) | 24.71 | 0.718 | 0.294 | 70.28 | 90.04 | 24.72 | 0.819 | 0.233 | 93.45 | 66.97 | 29.02 | 0.892 | 0.277 | 67.91 | 103.13 |

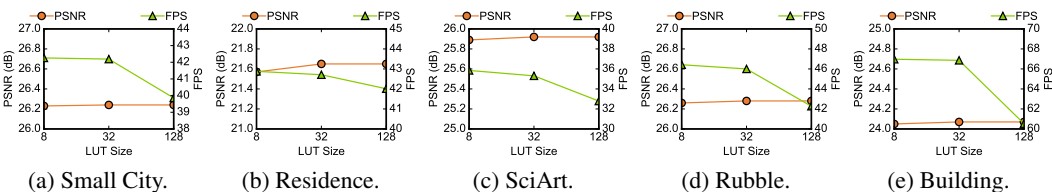

(a) Small City.     (b) Residence.     (c) SciArt.     (d) Rubble.     (e) Building.

Fig. 10: The sensitivity of rendering quality and performance to the lookup table size ($m$ intervals) in the rasterization stage. As $m$ increases, the performance decreases while the accuracy increases a little. We found that the size of 32 is a good trade-off across different datasets.

## 5.5 EVALUATION ON SMALL-SCALE DATASETS

We also evaluate VOYAGER on small-scale datasets in Tbl. 3. While rendering small-scale scenes is not the primary focus of this work, we include the results for completeness. We show that VOYAGER achieves relatively low rendering quality compared to the corresponding HierarchicalGS. The primary reason is that the LUT approximation introduces some artifacts compared to the exact $\alpha$-computation. Nevertheless, VOYAGER can achieve 2.5× and 3.4× speedup against HierarchicalGS and 3DGS, respectively. With some degree of quality sacrifices, VOYAGER at $\tau$ of 15 can further boost the performance to 3.6× and 4.8× speedup, respectively. This could be a potential limitation of VOYAGER on small-scale datasets, which we leave for future work.

## 5.6 SENSTIVITY STUDY

Fig. 10 shows the sensitivity study of rendering quality and the performance to the LUT size ($m$ intervals in Sec. 4.2), across five different scenes. We observe that increasing the LUT size from 8 to 32 results in notable improvements in rendering quality, while incurring only minimal performance overhead. However, further increasing the LUT size from 32 to 128 leads to a significant drop in performance, with negligible gains in visual quality. This trade-off suggests that a LUT size of 32 strikes a good balance between performance and rendering fidelity.

## 6 CONCLUSION

Human imagination is boundless. When harnessing Gaussians to construct virtual worlds, our rendering systems should be just as limitless. VOYAGER marks the first attempt toward enabling real-time city-scale Gaussian splatting on resource-constrained devices. By identifying and addressing the two primary bottlenecks, LoD search and splatting, we introduce two key innovations: temporal-aware LoD search and preemptive $\alpha$-filtering, both of which significantly enhance rendering efficiency. Overall, VOYAGER achieves up to 6.6× speedup in large-scale scenes and 4.8× speedup in small-scale scenes, all with comparable rendering quality.

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

## A    CODE OF ETHICS

I acknowledge that all co-authors of this work have read and commit to adhering to the ICLR Code of Ethics.

## B    LLM USAGE

We did not use LLM throughout the entire submission.

## C    SUPPLEMENTARY

### C.1    EXPERIMENTAL SETUP

**Hardware Platforms.**    We conduct both the performance and accuracy measurements on Nvidia Jetson Orin SoC with 5.325 TFLOPS (FP32) and 64 GB memory.

**Evaluation Metrics.**    Next, we describe how we obtain various performance and quality metrics.

- **Image Consistency (PSNR, SSIM, LPIPS).** To assess frame-to-frame image consistency, we compared generated videos by different acceleration methods against the original video results on image quality, using PSNR, SSIM, and LPIPS. PSNR is calculated using standard image processing libraries, e.g.,`skimage.metrics.peak_signal_noise_ratio`. Similarly, SSIM is calculated using `skimage.metrics.structural_similarity`. LPIPS is measured using the `lpips` library in Python.

- **Performance.** For performance, we report end-to-end latency, GPU memory consumption, and GPU energy consumption. For end-to-end latency, we measure the total time elapsed during one single frame rendering. Here, we use Python's `torch.cuda` module. We capture the start and end timestamps using `torch.cuda.event()` and use the difference between these two as the end-to-end latency. For GPU memory consumption, we use the built-in measurement to monitor the peak GPU memory usage during inference. The GPU power is directly obtained using the built-in power sensing circuitry on Orin.

Table 4: Efficiency evaluation of VOYAGER against the state-of-the-arts (Kerbl et al., 2023; Lee et al., 2024a; Liu et al., 2024; Ren et al., 2024; Kerbl et al., 2024). Here, we show the results on three large-scale datasets: HierarchicalGS (Kerbl et al., 2024), UrbanScene3D (Lin et al., 2022), and MegaNeRF (Turki et al., 2022).

| Quality Metrics | Methods | HierarchicalGS | UrbanScene3D | | | MegaNeRF | | |
|---|---|---|---|---|---|---|---|---|
| | | Small City | Residence | SciArt | Average | Rubble | Building | Average |
| FPS (Orin) ↑ | 3DGS | 12.18 | 13.61 | 9.59 | 11.60 | 12.28 | 11.78 | 12.03 |
| | CityGaussian | 24.86 | 18.47 | 20.00 | 19.24 | 25.20 | 15.59 | 20.40 |
| | OctreeGS | 13.55 | 32.63 | 27.97 | 30.30 | 31.24 | 26.85 | 29.04 |
| | HierarchicalGS ($\tau = 0.0$) | 8.95 | 13.02 | 11.70 | 12.36 | 12.70 | 16.97 | 14.83 |
| | HierarchicalGS ($\tau = 3.0$) | 11.43 | 12.93 | 12.44 | 12.69 | 13.25 | 17.52 | 15.39 |
| | HierarchicalGS ($\tau = 15.0$) | 15.53 | 17.25 | 18.63 | 17.94 | 16.79 | 22.46 | 19.63 |
| | VOYAGER ($\tau = 0.0$) | 28.70 | 42.24 | 32.63 | 37.43 | 43.55 | 63.65 | 53.60 |
| | VOYAGER ($\tau = 3.0$) | 42.19 | 42.71 | 35.31 | 39.01 | 46.0 | 66.86 | 56.42 |
| | VOYAGER ($\tau = 15.0$) | 68.84 | 58.19 | 59.15 | 58.67 | 64.24 | 94.52 | 79.38 |
| #Op. ($10^6$) ↓ | 3DGS | 732.69 | 2225.49 | 2863.02 | 2544.25 | 1407.22 | 1854.63 | 1630.92 |
| | CityGaussian | 649.68 | 2473.56 | 2571.05 | 2522.30 | 1177.23 | 3474.94 | 2326.09 |
| | OctreeGS | 2068.59 | 2088.34 | 2154.70 | 2121.52 | 1389.11 | 1008.29 | 1198.70 |
| | HierarchicalGS ($\tau = 0.0$) | 1086.99 | 2338.85 | 2946.90 | 2642.88 | 1993.58 | 1359.21 | 1676.39 |
| | HierarchicalGS ($\tau = 3.0$) | 857.64 | 2323.27 | 2681.50 | 2502.38 | 1948.93 | 1307.92 | 1628.42 |
| | HierarchicalGS ($\tau = 15.0$) | 591.01 | 1587.84 | 1506.68 | 1547.26 | 1368.09 | 996.03 | 1182.06 |
| | VOYAGER ($\tau = 0.0$) | 682.10 | 1205.48 | 1180.87 | 1193.17 | 580.58 | 522.95 | 551.77 |
| | VOYAGER ($\tau = 3.0$) | 584.15 | 1166.39 | 1071.29 | 1118.84 | 556.27 | 501.35 | 528.81 |
| | VOYAGER ($\tau = 15.0$) | 363.67 | 712.98 | 594.10 | 653.54 | 391.20 | 332.85 | 362.02 |
| Energy (mJ) ↓ | 3DGS | 443.52 | 396.78 | 562.99 | 479.88 | 423.37 | 407.54 | 415.46 |
| | CityGaussian | 209.16 | 292.30 | 260.01 | 276.16 | 190.47 | 333.49 | 261.98 |
| | OctreeGS | 398.54 | 207.37 | 165.49 | 186.43 | 160.04 | 193.69 | 176.87 |
| | HierarchicalGS ($\tau = 0.0$) | 581.16 | 399.37 | 444.44 | 421.90 | 409.61 | 306.37 | 357.99 |
| | HierarchicalGS ($\tau = 3.0$) | 455.03 | 402.08 | 417.85 | 409.96 | 392.37 | 296.73 | 344.55 |
| | HierarchicalGS ($\tau = 15.0$) | 405.95 | 378.15 | 371.99 | 375.07 | 372.24 | 273.68 | 322.96 |
| | VOYAGER ($\tau = 0.0$) | 188.13 | 127.85 | 165.50 | 146.67 | 123.99 | 84.83 | 104.41 |
| | VOYAGER ($\tau = 3.0$) | 127.99 | 126.44 | 152.94 | 139.69 | 117.47 | 80.76 | 99.12 |
| | VOYAGER ($\tau = 15.0$) | 78.44 | 92.80 | 91.29 | 92.05 | 84.06 | 57.13 | 70.60 |

Table 5: Quality evaluation of VOYAGER against the state-of-the-arts (Kerbl et al., 2023; Lee et al., 2024a; Liu et al., 2024; Ren et al., 2024; Kerbl et al., 2024). Here, we show the results on three large-scale datasets: HierarchicalGS (Kerbl et al., 2024), UrbanScene3D (Lin et al., 2022), and MegaNeRF (Turki et al., 2022). We do not report the result of *Campus* due to OOM on all methods.

| Quality Metrics | Methods | HierarchicalGS | UrbanScene3D | | | MegaNeRF | | |
|---|---|---|---|---|---|---|---|---|
| | | Small City | Residence | SciArt | Average | Rubble | Building | Average |
| PSNR (dB) ↑ | 3DGS | 23.32 | 21.86 | 21.43 | 21.65 | 25.68 | 20.25 | 22.96 |
| | CityGaussian | 22.01 | 22.03 | 21.43 | 21.73 | 25.59 | 22.96 | 24.27 |
| | OctreeGS | 21.33 | 21.77 | 22.18 | 21.97 | 25.56 | 23.55 | 24.56 |
| | HierarchicalGS ($\tau = 0.0$) | 26.34 | 21.69 | 25.95 | 23.82 | 26.47 | 24.15 | 25.31 |
| | HierarchicalGS ($\tau = 3.0$) | 26.24 | 21.65 | 25.92 | 23.79 | 26.29 | 24.07 | 25.18 |
| | HierarchicalGS ($\tau = 15.0$) | 25.39 | 21.11 | 25.11 | 23.11 | 23.88 | 22.75 | 23.31 |
| | VOYAGER ($\tau = 0.0$) | 26.34 | 21.69 | 25.95 | 23.82 | 26.47 | 24.15 | 25.31 |
| | VOYAGER ($\tau = 3.0$) | 26.24 | 21.65 | 25.92 | 23.78 | 26.28 | 24.07 | 25.17 |
| | VOYAGER ($\tau = 15.0$) | 25.06 | 20.67 | 24.64 | 22.66 | 23.43 | 22.51 | 22.97 |
| SSIM ↑ | 3DGS | 0.724 | 0.797 | 0.839 | 0.818 | 0.791 | 0.713 | 0.752 |
| | CityGaussian | 0.729 | 0.814 | 0.835 | 0.825 | 0.810 | 0.784 | 0.797 |
| | OctreeGS | 0.646 | 0.771 | 0.834 | 0.802 | 0.809 | 0.720 | 0.765 |
| | HierarchicalGS ($\tau = 0.0$) | 0.814 | 0.675 | 0.838 | 0.757 | 0.817 | 0.757 | 0.787 |
| | HierarchicalGS ($\tau = 3.0$) | 0.811 | 0.674 | 0.837 | 0.755 | 0.812 | 0.754 | 0.783 |
| | HierarchicalGS ($\tau = 15.0$) | 0.771 | 0.639 | 0.793 | 0.716 | 0.706 | 0.666 | 0.686 |
| | VOYAGER ($\tau = 0.0$) | 0.814 | 0.675 | 0.838 | 0.757 | 0.817 | 0.757 | 0.787 |
| | VOYAGER ($\tau = 3.0$) | 0.812 | 0.674 | 0.837 | 0.755 | 0.812 | 0.754 | 0.783 |
| | VOYAGER ($\tau = 15.0$) | 0.772 | 0.627 | 0.787 | 0.707 | 0.689 | 0.653 | 0.671 |
| LPIPS ↓ | 3DGS | 0.430 | 0.264 | 0.274 | 0.269 | 0.304 | 0.356 | 0.330 |
| | CityGaussian | 0.344 | 0.212 | 0.234 | 0.223 | 0.233 | 0.243 | 0.238 |
| | OctreeGS | 0.415 | 0.275 | 0.219 | 0.247 | 0.269 | 0.274 | 0.272 |
| | HierarchicalGS ($\tau = 0.0$) | 0.250 | 0.246 | 0.206 | 0.226 | 0.256 | 0.296 | 0.276 |
| | HierarchicalGS ($\tau = 3.0$) | 0.254 | 0.248 | 0.208 | 0.228 | 0.261 | 0.299 | 0.280 |
| | HierarchicalGS ($\tau = 15.0$) | 0.316 | 0.332 | 0.279 | 0.306 | 0.358 | 0.376 | 0.367 |
| | VOYAGER ($\tau = 0.0$) | 0.250 | 0.246 | 0.206 | 0.226 | 0.256 | 0.296 | 0.276 |
| | VOYAGER ($\tau = 3.0$) | 0.254 | 0.248 | 0.208 | 0.228 | 0.261 | 0.299 | 0.280 |
| | VOYAGER ($\tau = 15.0$) | 0.324 | 0.351 | 0.311 | 0.331 | 0.365 | 0.384 | 0.375 |

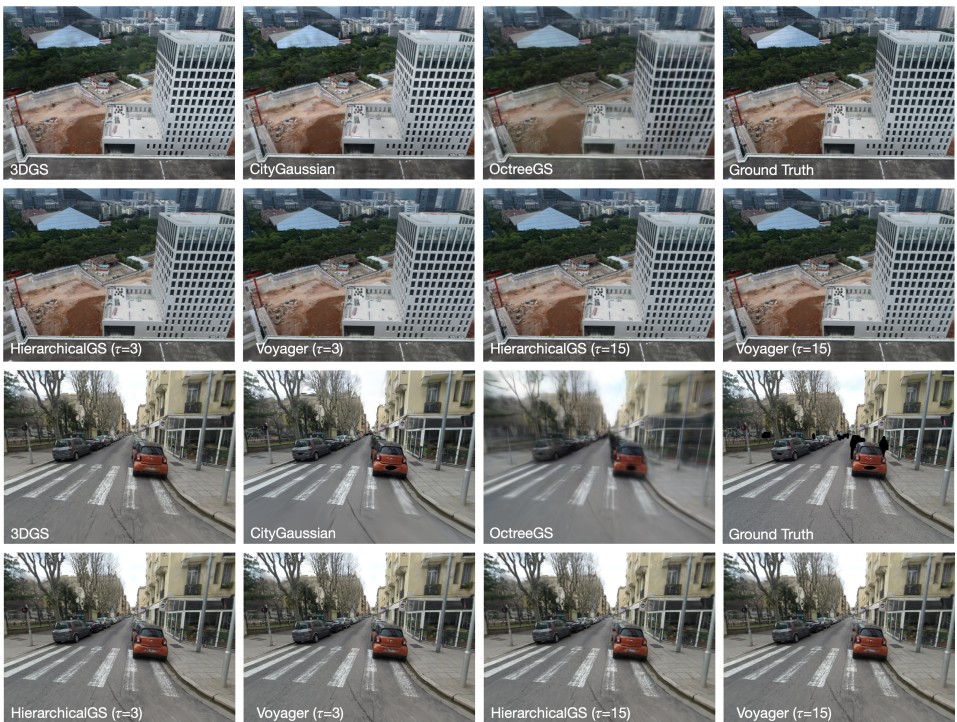

Fig. 11: Qualitative comparison of VOYAGER against other baselines Liu et al. (2024); Kerbl et al. (2024; 2023); Ren et al. (2024) on *SciArt* and *SmallCity*. **More results in the supplementary video.**

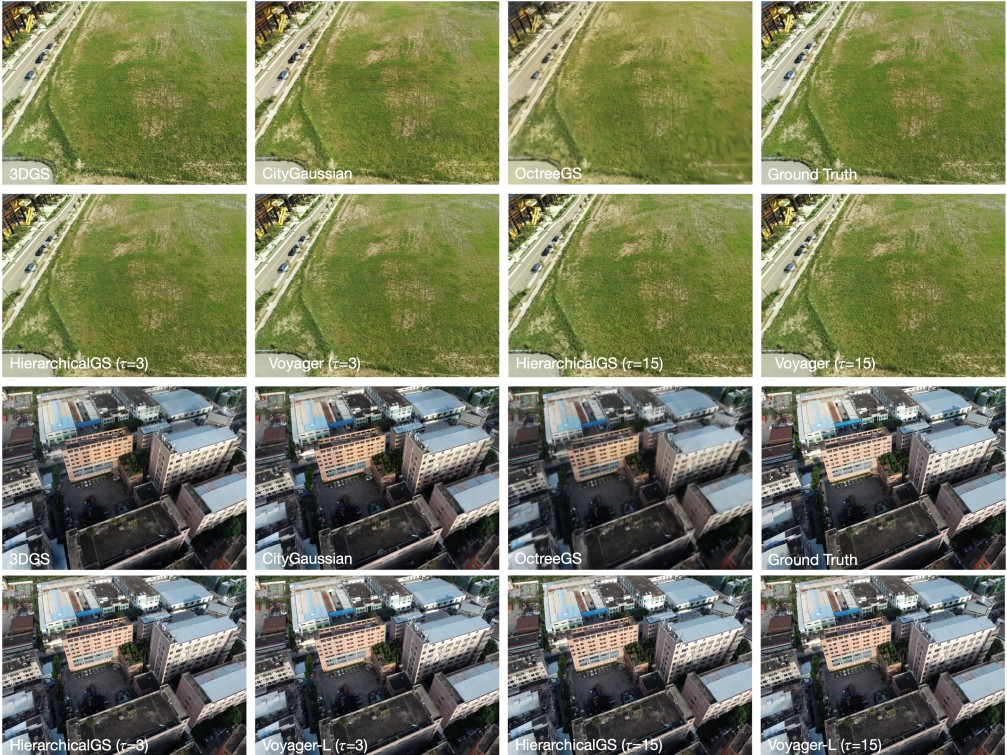

Fig. 12: Qualitative comparison of VOYAGER against other baselines Liu et al. (2024); Kerbl et al. (2024; 2023); Ren et al. (2024) on *Building* and *Residence*. **More results in the supplementary video.**

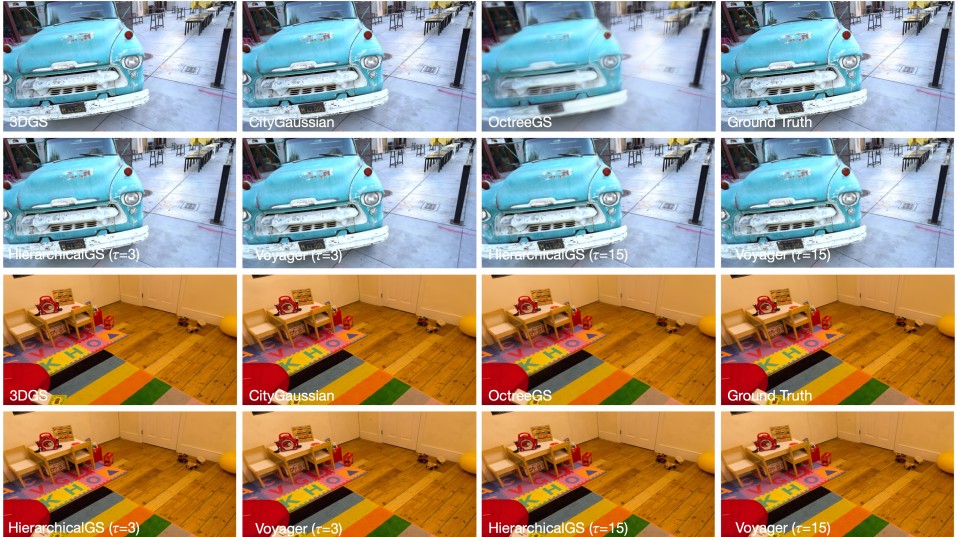

Fig. 13: Qualitative comparison of VOYAGER against other baselines Liu et al. (2024); Kerbl et al. (2024; 2023); Ren et al. (2024) on small-scale datasets. **More results in the supplementary video.**

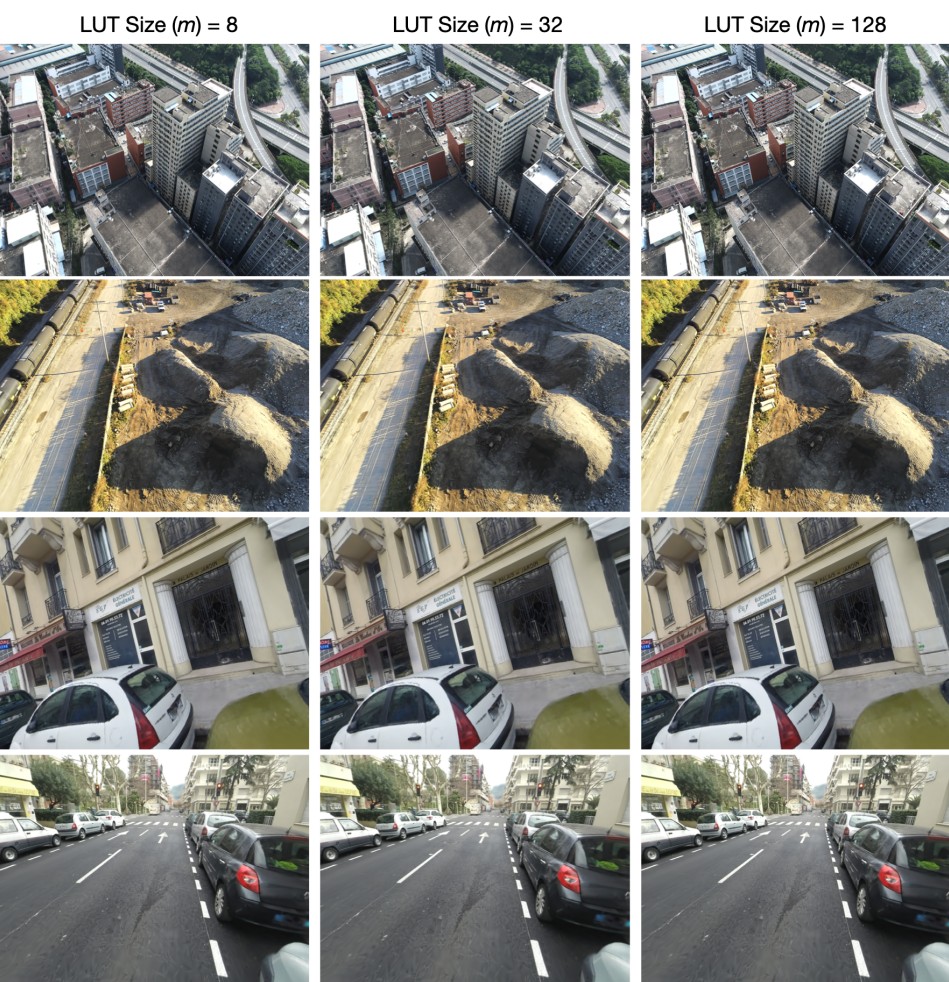

Fig. 14: Qualitative results of the sensitivity of rendering quality and performance to the lookup table size ($m$ intervals) in the rasterization stage.

