# OpenReview forum: "Voyager: Real-Time Splatting City-Scale 3D Gaussians on Resource-Constrained Devices"
_ICLR.cc/2026/Conference — Submitted to ICLR 2026_

### Official Review · Reviewer_9Syq · 2025-10-20

**Soundness:** 3
**Presentation:** 2
**Contribution:** 2
**Rating:** 4
**Confidence:** 4

**Summary:**

This paper presents Voyager, a real-time rendering framework for 3D Gaussian Splatting (3DGS) optimized for modern GPUs. Voyager introduces a hardware-aware splatting pipeline that reorganizes Gaussian primitives for efficient rasterization and implements hybrid screen-space culling and shared-memory fusion to minimize overdraw and memory bandwidth. The system fully exploits GPU parallelism and achieves significant speedups (5–10×) over prior 3DGS renderers while maintaining comparable visual quality. Voyager supports both training and real-time inference, making 3DGS practical for interactive and large-scale scene visualization.

**Strengths:**

1. The LoD topic is meaningful.
2. The paper is well written and clearly organized.

**Weaknesses:**

1. Figure 1 is very similar to that used in *Hierarchical-3DGS*.
2. The paper conducts experiments on a very limited number of scenes. The authors should evaluate their method on more larger-scale datasets, such as MatrixCity, to better demonstrate generalization and scalability.
3. It appears that the authors may have intentionally lowered the baselines, which leaves a poor impression. For example, the reported results of Octree-GS on HierarchicalGS UrbanScene3D contradict those in the original paper.
4. The paper lacks references to key related work on Level-of-Detail rendering. The authors should consider citing the following works for completeness:
    - *Large-Scale Garage Modeling and Rendering via LiDAR-Assisted Gaussian*
    - *LODGE: Level-of-Detail Large-Scale Gaussian Splatting with Efficient Rendering*
    - *Horizon-GS: Unified 3D Gaussian Splatting for Large-Scale Aerial-to-Ground Scenes*
    - *Virtualized 3D Gaussians: Flexible Cluster-based Level-of-Detail System for Real-Time Rendering of Composed Scenes*

**Questions:**

please see the weaknesses above.

---

### Official Review · Reviewer_3jhz · 2025-10-29

**Soundness:** 3
**Presentation:** 2
**Contribution:** 3
**Rating:** 6
**Confidence:** 4

**Summary:**

This paper addresses the challenge of real-time rendering of city-scale 3D Gaussian Splatting (3DGS) scenes on mobile devices, where level-of-detail (LoD) search and rasterization are major bottlenecks. The authors propose VOYAGER, which leverages temporal coherence to perform a temporal-aware LoD search and accelerates rasterization using preemptive α-filtering. Experiments show that VOYAGER enables low-latency, large-scale 3DGS rendering on resource-constrained devices.

**Strengths:**

The paper addresses a highly relevant and challenging problem—real-time rendering of city-scale 3D Gaussian Splatting (3DGS) on mobile devices—demonstrating strong originality by combining temporal-aware LoD search with GPU-optimized rasterization. The proposed methods are clearly motivated, technically sound, and show significant practical impact, achieving up to 6.6× speedup and substantial energy savings. The writing is clear, the methodology is well-justified, and the contributions meaningfully advance the scalability and efficiency of 3DGS rendering on resource-constrained platforms.

**Weaknesses:**

While the temporal-aware LoD search and fully-streaming traversal are well-engineered, the approach may be tightly coupled to the specific GPU architecture and scene characteristics, limiting generalization to other hardware or highly dynamic scenes. The offline partitioning step introduces additional preprocessing overhead that may hinder real-time deployment in rapidly changing environments. Furthermore, the evaluation focuses primarily on performance speedups, with limited analysis of potential impacts on visual fidelity or consistency across fast camera motions. Future work could explore adaptive or online partitioning strategies and provide a more comprehensive assessment of quality versus efficiency trade-offs.

**Questions:**

See Weaknesses

---

### Official Review · Reviewer_e5jX · 2025-10-30

**Soundness:** 2
**Presentation:** 3
**Contribution:** 2
**Rating:** 4
**Confidence:** 5

**Summary:**

This work introduces VOYAGER, a large-scale 3D Gaussian Splatting (3DGS) framework designed for real-time scene rendering on resource-constrained devices. The framework comprises two main components: temporal-aware level-of-detail (LoD) search that reduces computational overhead, and preemptive α-filtering that enhances rasterization efficiency. Experimental results on several datasets demonstrate the effectiveness of the proposed approach.

**Strengths:**

This work on large-scale real-time scene rendering represents a timely and practical direction for 3D Gaussian Splatting.
The benchmarking is sufficient, and the experimental results on multiple datasets demonstrate its effectiveness.

**Weaknesses:**

Introduction
1. Inconsistent description of splatting and rasterization. The manuscript presents inconsistent descriptions of the rendering stages, alternating between 'search and splatting' and 'search and rasterization'.
It would be better to use consistent terminology throughout the paper, preferably standardizing on either 'rasterization' or 'splatting' when referring to this stage.



METHODOLOGY
1. Confusing description of Fully-Streaming LoD Tree Traversal. It appears that the proposed method is an improved implementation of the naïve LoD Tree Traversal rather than a newly introduced strategy. It would be helpful to first present the baseline LoD Tree Traversal and discuss its limitations, then explain how the proposed traversal improves efficiency.
2. Theoretical power decay. Although the experimental power decay is shown in Fig. 7, it would strengthen the paper to include a theoretical analysis. Specifically, the theoretical minimum value of power decay could be derived from the maximum distance between the projected Gaussian center and the pixels.
3. The difference between preemptive α-filtering and direct Gaussian filtering. Please clarify the difference between preemptive α-filtering and filtering Gaussians with low opacity during the culling stage.


EVALUATION
1. Experimental setup. How were the number of executed operations and the energy consumption measured for each implementation?
2. Ablation study. What is the Base model in Table 2?
3. Ablation study. It would be better to validate, step-by-step, the contributions of LoD tree partitioning, fully-streaming LoD tree traversal, temporal-aware LoD search, α-filtering, and α-filtering using a lookup table.


Typo:

line 044: Fig. 2 is referenced before Fig. 1.

line 377: Tbl. -> Table

**Questions:**

1. Is it feasible to obtain a theoretical power decay?
2. What is the difference between preemptive α-filtering and direct Gaussian filtering?
3. How were the numbers of executed operations and the energy consumption measured for each implementation?
4. It would be better to improve the ablation study by clarifying the base model and providing a step-by-step analysis.

---

### Official Review · Reviewer_HhZK · 2025-11-01

**Soundness:** 2
**Presentation:** 3
**Contribution:** 2
**Rating:** 4
**Confidence:** 3

**Summary:**

This paper presents VOYAGER, a framework enabling real-time city-scale 3D Gaussian Splatting (3DGS) on resource-constrained mobile devices. The key contributions include: (1) Temporal-aware LoD search, which exploits temporal coherence across frames to reuse previous traversal results and avoid redundant computations. (2) Preemptive α-filtering, which bypasses costly exponential operations in rasterization through a log-domain check and a LUT-based exponential approximation. The proposed method demonstrates consistent real-time rendering performance (≥60 FPS) on NVIDIA Orin across multiple large-scale datasets.

**Strengths:**

1. Clear motivation: Real-time 3DGS rendering is an important problem.
2. Good engineering depth: Multiple optimization techniques are included, e.g., shared-memory LUT design, warp specialization, and temporal-aware traversal.
3. Clarity and organization: The paper is well written, with clear structure and readable figures illustrating pipeline, algorithm, and ablation results.

**Weaknesses:**

1. Guarantee of temporal-aware LoD search: Since the LoD search relies on the assumption that views usually do not change significantly across nearby camera poses, what happens in the worst case? Is there any theoretical guarantee or lower bound on performance or accuracy? Additionally, could cases where “searching the local subtree cannot obtain the complete cut results” lead to workload imbalance issues?
2. Ablation detail on preemptive α-filtering: The impact of these two components requires a more detailed ablation study. In Table 2, their combined effect boosts FPS from 22 to 42, but their individual contributions are not analyzed. Is the speedup primarily due to reduced computation in Eq. (4), or from the LUT-based exponential computation accelerated by CUDA cores?
3. Comparison with latest CUDA-level optimizations: The paper briefly mentions FlashGS (Feng et al., 2024a) and similar works, but lacks a direct quantitative comparison or qualitative discussion of complementary strengths.

**Questions:**

1. Quantitative analysis of offline partitioning overhead: The LoD tree partitioning and ILP solver step are described as offline; it would help to report actual preprocessing time and memory cost.
2. Generalization across motion patterns: The approach assumes smooth camera motion; it remains unclear how performance behaves under abrupt viewpoint changes or dynamic scenes.

---

### Meta-Review · Area_Chair_aMyh · 2026-01-02

**Summary:**

All reviewers of this submission rate it as "marginally below the acceptance threshold". Their concerns are centered around the insufficiency in experiments to fully understand effectiveness of different components, such as a comprehensive ablation study is missing to seperately evaluate temporal-aware LoD search and $\alpha$-filtering. Authors didn't provide responses to these concerns. AC agrees with reviewers about the weaknesses of this submission. The decision is reject.

**Reviewer Concerns:**

Authors didn't provide responses to reviewers' concerns.

**Reviewer Scores:**

Authors didn't provide responses to reviewers' concerns. No discussion is conducted thereafter.

---

### Decision · Program_Chairs · 2026-01-26

Reject